# Considerations for Applying Entropy Methods to Temporally Correlated Stochastic Datasets

**DOI:** 10.3390/e25020306

**Published:** 2023-02-07

**Authors:** Joshua Liddy, Michael Busa

**Affiliations:** 1Department of Kinesiology, University of Massachusetts Amherst, Amherst, MA 01003, USA; 2Institute for Applied Life Sciences, University of Massachusetts Amherst, Amherst, MA 01003, USA

**Keywords:** entropy, biomechanics, human movement, temporal correlations, sample entropy, information theory, dynamical systems theory

## Abstract

The goal of this paper is to highlight considerations and provide recommendations for analytical issues that arise when applying entropy methods, specifically Sample Entropy (SampEn), to temporally correlated stochastic datasets, which are representative of a broad range of biomechanical and physiological variables. To simulate a variety of processes encountered in biomechanical applications, autoregressive fractionally integrated moving averaged (ARFIMA) models were used to produce temporally correlated data spanning the fractional Gaussian noise/fractional Brownian motion model. We then applied ARFIMA modeling and SampEn to the datasets to quantify the temporal correlations and regularity of the simulated datasets. We demonstrate the use of ARFIMA modeling for estimating temporal correlation properties and classifying stochastic datasets as stationary or nonstationary. We then leverage ARFIMA modeling to improve the effectiveness of data cleaning procedures and mitigate the influence of outliers on SampEn estimates. We also emphasize the limitations of SampEn to distinguish among stochastic datasets and suggest the use of complementary measures to better characterize the dynamics of biomechanical variables. Finally, we demonstrate that parameter normalization is not an effective procedure for increasing the interoperability of SampEn estimates, at least not for entirely stochastic datasets.

## 1. Introduction

The biomechanics of human movement are characterized by time-varying behavior that is amenable to analysis using entropy methods, which are grounded in principles of information theory and dynamical systems theory. Recently, entropy methods have received considerable attention from researchers and clinicians interested in human movement [1,2,3,4,5,6,7,8,9,10,11,12,13,14,15,16,17]. Despite this growing interest, it is not always clear when and how to leverage these powerful analytical tools while avoiding misapplication and misinterpretation.

The goal of this paper is to highlight considerations and provide recommendations for analytical issues associated with applying entropy methods to temporally correlated stochastic datasets, which are representative of a broad range of biomechanical variables, including spatiotemporal gait parameters and center-of-pressure trajectories. Before addressing specific problems and goals, we provide a brief overview of information theory and the concept of information, entropy as a measure of information, entropy methods for physiological datasets, and uses of entropy methods in biomechanical applications.

### 1.1. Information Theory and Information

Information theory is a branch of mathematics concerned with probability and statistics that is broadly applicable to probabilistic sequences of measurements [18]. In information theory, information is related to the number of responses a system produces and the associated probabilities of those responses under fixed conditions. Information is closely related to uncertainty; the more uncertain the outcomes, the more information gained from new observations, and vice versa.

The information produced by a system can change from one setting to another when the number or probability of the responses changes, such as when environmental conditions or external constraints are varied. Information refers to how the system could have behaved, not how it did, and reflects the degree of constraint on the responses [19]. Systems characterized by more information can produce more flexible responses (i.e., there is greater use of the response space or more uncertainty about future responses), whereas systems characterized by less information produce more rigid responses (i.e., there is reduced use of the response space or less uncertainty about future responses).

### 1.2. Entropy and Information

Information can be defined formally from a set of potential responses and their associated probabilities:(1)$$H=-\underset{i}{\sum }p_{i}\log_{2}p_{i},$$
where $p_{i}$ is the probability associated with the *i*th response. This definition, which is known as Shannon entropy, is the foundation of modern information theory. Shannon entropy has the property that information is zero when the response is known and maximal when all responses occur with the same likelihood. Intuitively, information increases as the number of possible responses grows and the probabilities remain uniform. Another way to think about information is that it represents the surprise associated with a response—i.e., we are least surprised when the response is known in advance and most surprised when all responses are equally likely.

Shannon entropy measures the average information associated with a static probability distribution, whereas its dynamical counterparts, such as Kolmogorov–Sinai (KS) entropy [20,21], measure the rate of information generation from a deterministic dynamical system. However, KS entropy is not applicable to the finite, noisy datasets commonly measured from physiological systems [22]. To overcome this limitation, several related methods were developed to accommodate datasets obtained from systems with deterministic and stochastic components.

### 1.3. Entropy Methods for Physiological Data

Modeling the processes that generate physiological datasets is challenging. Entropy methods are model-free analytical tools for quantifying the degree of randomness in a dataset without making assumptions about the sources that generated it. Importantly, entropy methods measure randomness in the temporal organization of a dataset independent of its associated moment statistics [23].

Approximate Entropy (ApEn) was developed to quantify regularity in short, noisy physiological datasets [24]. For stochastic processes, K-S entropy returns infinite values, whereas ApEn returns finite values, allowing stochastic datasets to be compared [25]. ApEn quantifies the likelihood that sequences of a fixed template length (*m*) that are considered close within a tolerance (*r*) remain close when the template length is increased (*m* + 1) for a dataset of *N* observations. ApEn was subsequently used in a wide range of clinical applications, such as distinguishing age- and disease-related changes to cardiovascular dynamics [26,27].

ApEn has a self-matching bias and lacks relative consistency such that the pattern of results is not always maintained when hyperparameters are changed [25,28,29,30]. Consequently, Sample Entropy (SampEn) was developed to eliminate self-matching and the template-wise approach for computing probabilities [28]. SampEn was shown to be more accurate and reliable than ApEn, and serves as the basis for other entropy techniques, such as Control Entropy [31,32] and multiscale entropy (MSE) [33,34]. A modified definition of SampEn accounts for temporal correlations among nearby template vectors, which can occur due to oversampling, by including a time delay, τ [35].

Because this paper examines temporally correlated stochastic datasets, SampEn is a more suitable method than ApEn. Stochastic processes exhibit weak dependence compared to deterministic processes, creating diffuse dynamics such that nearly all partitions contain near-zero probabilities [29]. The self-matching bias of ApEn is particularly problematic in this case because the conditional probability log(A/B) can be much different than log((A + 1)/(B + 1)). As mentioned before, SampEn also serves as the basis for other entropy algorithms, such as MSE and Control Entropy. Thus, we restrict our focus to SampEn.

### 1.4. Entropy Methods in Biomechanical Applications

Entropy methods, such as ApEn and SampEn, have been widely used in clinical applications to examine changes to the regularity of physiological data, including cardiovascular [26,27], respiratory [36,37,38], and neural dynamics [39,40,41]. More recently, entropy methods have gained traction in the biomechanics community and provided important insights into human postural control, locomotion, and muscle activity.

Entropy measures have been applied to standing posture to examine how changes to the center of pressure dynamics support precision manual tasks [1], understand how interactions between aging and attentional states impact balance [2], as well as characterize short-term changes from training [3] and long-term changes from neurological disease progression [4]. Entropy has also been applied to gait dynamics to evaluate fall risk [5,6] and identify age-related changes to dual-task function [7,8]. Further insights into age- and disease-related changes to postural control and gait have been gained from MSE approaches [9,10,11,12,13].

SampEn has also been applied to electromyographic records to distinguish voluntary muscle activity from background noise [14], identify the onset of muscle fatigue [15,16], and improve pattern recognition to guide prosthetic limb control [17]. As the use of entropy methods in biomechanical applications grows, it is important to highlight common analytical challenges and identify solutions to facilitate accurate and reliable estimation, improve interpretability, and increase interoperability between datasets.

### 1.5. Problem Statement

In this paper, we examine problems related to the application of entropy methods to temporally correlated stochastic data, which adequately describe many biomechanical variables, such as spatiotemporal gait parameters. Stochastic datasets are often derived from event-based series—e.g., stride time is commonly measured as the interval between ipsilateral heel strikes. With this type of data, the goal is to understand cycle-to-cycle changes rather than within-cycle changes that can be derived from continuous measurements, such as joint angles [42], which have stronger deterministic characteristics. Below, we consider three problems that arise when working with stochastic datasets.

First, past research has examined the effect of transient spikes or outliers on entropy measures [43,44,45]. More deterministic signals are robust to outliers, and entropy estimates increase only slightly as the number and magnitude of outliers increases [44]. By contrast, stochastic signals are more sensitive to outliers because the standard deviation becomes inflated when the scale of the signal is small relative to the outliers. This makes the radius of similarity, *r*, large relative to the ‘true’ signal dynamics, which reduces entropy estimates because there are more template matches. Thus, lower entropy estimates can result both from increased regularity or outliers [45]. Despite the importance of these observations, recommendations for handling outliers when working with stochastic datasets are lacking. Should *r* be adjusted to reduce the bias? Should outliers be removed? More importantly, how do we strike a balance between minimizing bias and preserving the underlying dynamics?

Second, rather than representing uncorrelated noise, stochastic datasets often contain temporal correlations characterized by increases in the statistical likelihood that subsequent variations occur in the same (persistence) or opposite (anti-persistence) direction, which should intuitively be characterized by greater regularity (i.e., lower entropy). The formalization adopted here derives from two related families of correlated stochastic processes: fractional Gaussian noises (fGn) and fractional Brownian motions (fBm) [46]. The fGn are stationary, while the fBm are nonstationary. Both fGn and fBm are described by the Hurst exponent (*H*), which characterizes temporal correlation properties ranging from anti-persistent to persistent on the interval ]0,1[. The cumulative sum of an fGn produces a corresponding fBm with the same *H*, while the difference of an fBm produces a corresponding fGn. One problem is that distinct temporal correlation properties may be characterized by similar entropy estimates because persistence and anti-persistence create dynamical regularities, making it harder to distinguish potentially meaningful differences between experimental conditions or populations.

Third, another challenge with entropy methods is the limited interoperability of results obtained from independent studies, as others have indicated [47]. Interoperability refers to the ability to access data from multiple sources without loss of meaning and integrate data for analytical purposes, such as making comparisons or generating predictions. Here, we focus on one aspect of interoperability: parameter normalization. Entropy estimates have a lower bound of zero and an upper bound that is determined by multiple hyperparameters. While SampEn is formally independent of dataset length, estimates improve with greater numbers of observations because the conditional probabilities are better approximated [28]. Differences in dataset length may therefore produce differences in SampEn obtained from independent studies, which could be remedied by normalizing parameter estimates.

### 1.6. Goal Statement and Contributions

The goal of this paper is to provide recommendations to address analytical issues associated with the application of entropy methods, specifically SampEn, to stochastic datasets that are commonly encountered in biomechanical applications. Throughout the paper, we emphasize the importance of leveraging multiple techniques to better describe the dynamics of stochastic datasets. Where applicable, we highlight potential workarounds—some of which have been examined in previous work, others for which challenges remain. Overall, this paper provides a basic framework for applying SampEn to temporally correlated stochastic datasets, which represent a wide range of biomechanical processes, with considerations of issues ranging from process characterization to data cleaning to interpretation of results.

Specifically, we examine the use of autoregressive fractionally integrated moving average (ARFIMA) modeling to estimate the temporal correlation properties of stochastic datasets and classify datasets as stationary or nonstationary. The rationale for using simulated datasets rather than experimental datasets is that the temporal correlation properties are known in advance, making it possible to systematically assess the accuracy and reliability of SampEn across a range of input signals. ARFIMA modeling is then used to aid in data cleaning procedures and reduce biases in SampEn estimates due to the presence of outliers. We then demonstrate the challenge of discriminating the temporal correlation properties of stochastic datasets when using SampEn alone, leading to the suggestion of a multiple method approach. Finally, we demonstrate that parameter normalization is not an effective procedure for increasing interoperability, at least not for fully stochastic datasets.

## 2. Materials and Methods

### 2.1. Simulations

To produce simulated data with a range of temporal correlation properties, we used ARFIMA modeling following the methods described in [48], which demonstrated that simulations produced via ARFIMA modeling were more accurate than those generated by the Davies–Harte algorithm [49] and spectral synthesis method [50]. ARFIMA models are an extension of autoregressive integrated moving average (ARIMA) models, which are useful for understanding short-term correlations in stochastic data [51]. ARIMA models have three parameters: the autoregressive order (*p*), the moving average order (*q*), and the differencing order (*d*), which indicates how many times the data need to be differenced to be considered stationary.

ARIMA (*p*, *d*, *q*) models can be used to simulate data with specified parameters. By allowing *d* to take on fractional values, ARIMA models can produce long-term dependence, leading to the more general ARFIMA models [52]. ARFIMA models are only applicable to stationary data with d∈]−0.5, 0.5[. Conveniently, *d* is linearly related to the scaling exponent (α) in the fGn/fBm model [48]:(2)$$\alpha =\frac{2d+1}{2}$$
where α∈ ]0,1[ for fGn and α∈ ]1,2[ for fBm. For fGn α = *H* and for fBm α = *H* + 1. This relation was exploited to create stationary and nonstationary stochastic datasets.

ARFIMA simulations have been shown to generate accurate and reliable datasets, independent of the method used for parameter estimation [48]. To create the simulated data, we used the freely available *ARFIMA simulations* library for MATLAB [53]. Only ARFIMA (*0*, *d*, *0*) models, which omit the *p* and *q* parameters responsible for short-term correlations, were used. These models produce stationary fGn processes with d∈]−0.5, 0.5[, which can be cumulatively summed to obtain nonstationary fBm processes.

We created simulations with expected α values of 0.1 to 1.9 in increments of 0.1 (Table 1). One independent set of simulations was obtained for fGn simulations with 0.1 ≤ α ≤ 0.9. Another independent set of simulations was cumulatively summed to obtain fBm simulations with 1.1 ≤ α ≤ 1.9. Additionally, to obtain α ≅ 1, we created simulations with α = 0.99. There were 100 simulations for each expected α value. We also examined three dataset lengths (*N* = 250, 500, and 1000). Each dataset length was produced independently—i.e., shorter datasets were not cropped from longer ones. The mean and standard deviation of each dataset was 0 and 1, respectively.

### 2.2. ARFIMA Modeling

To estimate *d*, ARFIMA (*0*, *d*, *0*) models were fit to the simulated data using the Whittle approximation in the freely available *ARFIMA(p,d,q) estimator* library for MATLAB [54], which also requires the MFE toolbox [55]. Because ARFIMA models are restricted to stationary processes and produce bounded estimates of *d*, the Whittle approximation may not converge. Following recommendations by [56] and procedures adopted by [48], nonstationary processes were differenced, and the model fit to the stationary increments. To classify data as nonstationary, the model was fit before differencing. When the algorithm failed to converge, the data were considered nonstationary, and the model was reapplied to the differenced data. If the data were differenced, the *d* estimate was adjusted by adding 1 and then converted to α (Equation (2)).

Detrended fluctuation analysis (DFA) has been suggested for classifying stochastic datasets as stationary or nonstationary (e.g., [57]). Temporal correlations were separately estimated using evenly spaced average DFA [58,59] with the following hyperparameters: *n*_min_ = 10; *n*_max_ = *N*/4; *k* = 26, 37, and 47 for the three dataset lengths, respectively; and linear detrending. The decision to use ARFIMA models to measure temporal correlations was supported by observations of comparatively lower biases and standard deviations, which is consistent with previous findings [48]. ARFIMA is both a better estimator (Appendix A) and classifier (Appendix A) of the stochastic processes considered in this paper. A further benefit of the ARFIMA approach is that it does not require hyperparameter selection, which reduces the need to make decisions about selecting appropriate values.

### 2.3. SampEn(m, r, τ)

SampEn(*m*, *r*) has two hyperparameters: the template length, *m*, and the radius of similarity, *r* [28]. Both *m* and *r* need to be specified. SampEn is estimated as the negative logarithm of the conditional probability (*A*/*B*) that sequences of *m* observations that are close within tolerance *r* (*B*) remain close when the sequence is increased to *m* + 1 observations (*A*). SampEn eliminates the self-matching bias of ApEn by comparing each template vector to all other template vectors except itself. Furthermore, SampEn computes the conditional probability *A*/*B* for the entire sequence of observations instead of template-wise, which avoids the requirement that each template vector has at least one match. Note: SampEn(*m*, *r*) is a parameter, while SampEn(*m*, *r*, *N*) is a statistic estimated from a dataset.

To select *m*, different methods have been proposed, such as fitting autoregressive models [28,43,45]. However, stochastic datasets, such as those considered in this paper, are characterized by a wide range of temporal correlation properties, which would lead to the selection of different *m* for different ranges of α. We selected *m* = 1 based on the observation that the precision of SampEn estimates decreased for larger *m* (Appendix A).

To select *r*, different methods have also been proposed, such as minimizing the relative error in the conditional probability *A*/*B* or SampEn(*m*, *r*, *N*) [43,45]. We selected *r* = 0.25 based on the observation that the precision of SampEn was worse for *r* < 0.25 and only marginally improved for *r* > 0.25 (Appendix A). This decision was made to balance robust estimation against reducing the ability to discriminate distinct temporal sequences when large *r* values are selected.

A more general definition of SampEn includes a third hyperparameter: the time delay, τ [35]. The time delay, τ, represents the lag between successive elements in the template vectors. Compared to SampEn(*m*, *r*), where τ = 1 by default, SampEn(*m*, *r*, τ) produces more accurate estimates when data are oversampled or contain temporal correlations [35,60]. We used τ = 1 for all analyses in this paper. τ can be estimated using the autocorrelation function or average mutual information [61]. For α ≤ 0.5, the autocorrelation function decays exponentially, yielding τ = 1. For 0.5  ≤ α ≤ 1, the autocorrelation function decays more slowly and τ will be greater than 1; τ is even larger for nonstationary processes (α > 1). SampEn estimates obtained from stochastic processes containing long-term correlations (α > 5) were relatively unaffected by different τ values (Appendix A). Thus, τ appears to be less influential when using SampEn to interrogate stochastic datasets. However, τ will likely play a more important role when working experimental datasets that represent a mixture of deterministic and stochastic components.

### 2.4. Outlier Generation

To examine the impact of extreme observations on SampEn estimates, we obtained baseline measures of SampEn(*m* = 1, *r* = 0.25, τ = 1) for each of the simulated datasets. We then adjusted a small proportion of the observations (1%) by adding values drawn from a standard normal Gaussian distribution multiplied by three times the peak-to-peak signal amplitude, the same as previous work [44]. SampEn(*m* = 1, *r* = 0.25, τ = 1) was then applied to the contaminated datasets to quantify biases in parameter estimates.

### 2.5. Parameter Normalization

To create interoperable parameter estimates for SampEn, we examined a normalization procedure based on the maximum entropy attainable for a dataset. SampEn parameter estimates have a lower bound of 0 and an upper bound of ln(*N* − *m* − *1*) + ln(*N* − *m*) – ln(2) [28]. The lower bound is reached when all template vectors match—i.e., the conditional probability *A/B* = 1, which is unlikely in experimental data. The upper bound is defined for SampEn(*m*, *r*) and explicitly depends on the dataset length, *N*, and the template length, *m.* This value is approached when only small numbers of template matches are found. However, a more general expression can be derived for SampEn(*m*, *r*, τ), which includes the time delay, τ:(3)ln(N−mτ−1)+ln(N−mτ)−ln(2).

The two expressions are equivalent when τ = 1, which is used throughout this paper. To normalize the entropy estimates, we divided each value by its upper bound—i.e., the maximum entropy value. This may facilitate comparisons of estimates obtained for datasets with different lengths. For example, the upper bounds examined in this study were 10.32, 11.72, and 13.12 for the short (*N* = 250), medium (*N* = 500), and long (*N* = 1000) datasets, respectively. By normalizing the entropy estimates from 0 to 1, it becomes possible to compare datasets of different lengths.

To examine the effect of normalization, we generated simulations with two additional dataset lengths that exceeded those primarily considered in this paper. The longer dataset lengths were *N* = 5000 and 10,000. Both dataset lengths are quite large compared to the datasets commonly collected in laboratory-based experiments or clinical settings. However, with the growing availability and portability of mobile health monitoring technologies, increasingly large datasets are becoming feasible to obtain.

## 3. Results

### 3.1. Estimation of Temporal Correlations Using ARFIMA Modeling

To verify that the simulated data were characterized by the expected temporal correlation properties, ARFIMA (*0*, *d*, *0*) models were fit and the *d* estimates were converted to the scaling exponent, α. We then quantified the bias and standard deviation of the estimates, which are presented in Table 2. The short datasets consistently underestimated entropy (Figure 1A), whereas the longer datasets were unbiased except for values close to α = 2 (Figure 1B,C). The standard deviation was the greatest for the short datasets and was elevated for α > 1.5. This trend was reduced in the longer datasets. In summary, the ARFIMA simulated datasets contained reliably different temporal correlation properties ranging from anti-persistent stationary data (e.g., α < 0.5) to persistent nonstationary data (e.g., α > 1.5), as characterized by ARFIMA (0, *d*, 0) models.

### 3.2. Classification of Stochastic Data Using ARFIMA Modeling

To examine the use of ARFIMA modeling for data classification, we computed the proportion of the 100 simulations classified as nonstationary for each of the expected α values. An ideal classifier would indicate that data with 0 < α < 1 were stationary and data with 1 < α < 2 were nonstationary. In practice, approximately 50% of the simulations close to the 1/f boundary (i.e., α = 1) were classified as nonstationary (Figure 2A). This was consistent across dataset lengths. Shorter datasets were more prone to misclassification near the boundary, but the error rates were less than 5% for all dataset lengths.

Classification errors decreased the farther the expected α value was from the 1/f boundary. However, classification errors were observed for 0.8 < α < 1.2. To better characterize the classification accuracy of the ARFIMA model, we conducted a follow-up analysis by generating simulated datasets with temporal correlations varying from 0.7 < α < 1.3 in increments 0.01. The same three dataset lengths were examined. We generated 500 simulations for each combination of α and *N*, leading to a total of 90,000 independent datasets. ARFIMA (0, *d*, 0) models were again used to obtain *d* estimates, which were subsequently converted to α and used to assess classification errors.

The follow-up analysis revealed differences in classification accuracy across the dataset lengths (Figure 2B). Short datasets (*N* = 250) were classified least accurately on both sides of the 1/f boundary with error rates exceeding 5% from 0.90 < α < 1.09. This was not unexpected because the standard deviation of the α estimates was elevated compared to the longer datasets. Medium datasets (*N* = 500) were classified less accurately than long datasets (*N* = 1000) below the boundary, with errors exceeding 5% at α = 0.92 and α = 0.95, respectively. By contrast, classification accuracy was comparable between the medium and long datasets above the boundary, with error rates exceeding 5% at α = 1.05. This asymmetry is likely due to the increased standard deviation of the α estimates below compared to above the boundary for the medium datasets.

In summary, ARFIMA modeling is also an accurate and reliable method for classifying whether stochastic datasets are stationary or nonstationary. Classification accuracy is dependent on dataset length near the 1/f boundary (α = 1.0) and improves with more data. In the following section, we demonstrate the utility of this approach for improving entropy estimates following data cleaning procedures.

### 3.3. Outlier Removal with Data Classification Reduces Biases in Entropy Estimates

The main problem with outliers is that they inflate the standard deviation. Because data are commonly normalized before applying SampEn (e.g., [43]), the amplitude of the ‘true’ data will be reduced, which will increase the likelihood of detecting template matches. The extent to which the standard deviation changes depends on the number and magnitude of the outliers. Datasets were manipulated by adjusting a small proportion of observations (1%) with peak magnitudes up to 3× the range of the dataset.

Entropy estimates were substantially lower in the presence of outliers (Figure 3A–C). Biases were relatively consistent across dataset length and greater than 25% in most cases. However, lower biases were observed for nonstationary datasets with persistent increments (i.e., α > 1.5). The standard deviation of the bias decreased with increasing dataset length, possibly because more observations were manipulated in the longer datasets. Thus, a small number of outliers had a disproportionate impact on entropy estimates, consistent with previous findings [44]. 

One potential solution is to adjust *r* because the amplitude of the ‘true’ data is reduced when the data are normalized with outliers. An adjustment can be applied to *r* by taking the ratio of the standard deviation without outliers to the standard deviation with outliers. For example, if *r* = 0.2 and the ratio of the standard deviations is 0.5, then the adjusted *r* is 0.5 *r* or 0.1. This creates stricter requirements for identifying template matches and improves entropy estimates. To examine the efficacy of this procedure, outliers were identified using a highly conservative threshold of five times the median absolute deviation [62]. The adjusted *r* values were computed and SampEn was reapplied to the outlier-contaminated datasets.

Adjusting *r* substantially reduced the bias for all datasets lengths (Figure 3D–F). The bias was less than 5% and the standard deviation less than 3% for α < 1.5. Positive biases were observed for α ≥ 1.5. The biases increased as α and *N* increased. Similar trends were observed for the standard deviation of the bias. These findings indicate that adjusting *r* reduces the influence of outliers on entropy estimates, particularly for stationary datasets; however, this procedure does not consistently work for nonstationary datasets.

Another potential solution is to remove outliers without adjusting *r*. In practice, this is a simple procedure but one that requires caution. Here, we assumed that observations flagged as outliers were not representative and should be removed. The maximum number of possible outliers was 3, 5, and 10 for the short, medium, and long datasets, respectively. The conservative threshold of five median absolute deviations detected less than or equal to the maximum possible number of outliers for nearly all datasets. False positives were detected in only 2 of the 5,700 total datasets, demonstrating the conservative nature of this approach.

Removing outliers further reduced the bias for all dataset lengths (Figure 3G–I). The biases were less than 1% and standard deviation less than 2% for α < 1.5. Short datasets showed the greatest variance within this range. Again, positive biases were observed for α ≥ 1.5—although the magnitudes were smaller than when adjusting *r*. This indicates that outlier removal is an effective procedure for minimizing the impact of outliers on entropy estimates. However, similar to adjusting *r*, outlier removal is not effective for nonstationary datasets with persistent increments (α > 1.5).

To address this limitation, ARFIMA models were applied to the data to classify datasets as stationary or nonstationary. One challenge with identifying outliers from nonstationary datasets is that they do not stand out. However, if the data are differenced, outliers can be easily identified and removed. This procedure was applied to the data and compared to outlier removal without classification. Data classification using ARFIMA modeling reduced biases across the entire α range independent of dataset length (Figure 4). The maximum biases were 1.5%, 3.5%, and 3.1% for the short, medium, and long datasets. However, this method produced some tradeoffs. While the standard deviation of the bias was reduced for the nonstationary datasets with persistent increments (α > 1.5), it increased slightly for nonstationary datasets with anti-persistent increments (α < 1.5).

To summarize, when stochastic datasets contain potential outliers, we recommend using a scaled median absolute deviation to identify and remove those observations. The scale factor should be tuned so that the number of outliers is acceptably small. This procedure was effective for stationary datasets (α < 1), but less so for nonstationary datasets (α > 1). Consequently, ARFIMA modeling was used to classify data as stationary or nonstationary before applying the outlier removal procedure, which substantially reduced biases in entropy estimates with minor tradeoffs in the standard deviation of the bias.

### 3.4. Problems with Discriminating Temporally Correlated Stochastic Datasets with SampEn

Because stochastic datasets commonly contain temporal correlations, entropy should decrease for both anti-persistent and persistent processes, as well as nonstationary ones, leading to potentially equivalent outcomes despite being characterized by distinctly different temporal correlation properties. We applied SampEn(*m* = 1, *r* = 0.25, τ = 1) to the simulated datasets with the following expectations: (1) entropy should be greatest for uncorrelated time series (α = 0.5) and (2) entropy will decrease as temporal correlations increase (α ≠ 0.5). This should produce an inverted parabolic relationship between entropy measured by SampEn and *α*. Greater numbers of data points were not expected to impact these predictions.

The estimated and expected α values obtained from ARFIMA modeling showed a linear relationship with a slope close to 1 (Figure 5A–C). Biases and standard deviations decreased with increasing dataset length. Thus, ARFIMA modeling accurately characterizes the temporal correlation properties of stationary and nonstationary datasets independent of dataset length, which is consistent with past findings [48].

By contrast, SampEn estimates were characterized by an inverted parabolic relationship across the expected *α* values independent of dataset length (Figure 5D–F). Stationary datasets containing temporal correlations—i.e., anti-persistent (α < 0.5) and persistent (α > 0.5) processes—were characterized by lower entropy compared to uncorrelated noise (α = 0.5), as expected. The only exception was the short datasets, where α = 0.4 was slightly larger than α = 0.5. Strongly persistent processes, which contain long-term correlations, were characterized by greater regularity than strongly anti-persistent processes, leading to a right-skewed asymmetric relationship between entropy and α. Means and standard deviations are reported in Table 3.

Nonstationary datasets were characterized by decreasing regularity for 1 < α < 2. This makes intuitive sense because these processes become more regular as the increments shift from anti-persistent to uncorrelated to persistent. The precision of SampEn was particularly poor for the nonstationary anti-persistent processes (1 < α < 1.5). Greater precision was observed for the nonstationary persistent processes (1.5 < α < 2), likely because SampEn has a lower bound of zero. Again, means and standard deviations are reported in Table 3.

The nonlinear relationship between SampEn and α suggests the need to use complementary measures when examining stochastic datasets. Specifically, we suggest the use of ARFIMA modeling or other methods capable of estimating temporal correlations. Otherwise, SampEn will have difficulty distinguishing between stationary anti-persistent and persistent processes, as well as identifying subtle differences between nonstationary persistent processes, both of which may indicate relevant differences between experimental conditions or populations.

### 3.5. Normalization of SampEn Estimates across Different Dataset Lengths

SampEn is bounded from below by ln(1) = 0, which occurs when the conditional probability *A/B* = 1, and from above by ln(*N* − *m*τ − *1*) + ln(*N* − *m*τ) – ln(2) Equation (3). The upper bound is determined by multiple hyperparameters but is principally influenced by the dataset length, *N*. This value represents the maximum value possible for specific hyperparameter combinations. Although the radius of similarity, *r*, is not explicitly included, it defines how close sequences of observations must be to be considered a match. Thus, SampEn estimates will grow with decreasing *r*.

In practice, the estimate will not approach the maximum value because *r* is relatively large—commonly close to 0.2 standard deviations—such that the number of matches will also be relatively large. To improve the interoperability of SampEn estimates, the obtained estimate can be normalized by its maximum value, which transforms it to a 0 to 1 scale, where 0 corresponds to complete regularity and 1 corresponds to complete randomness. We now consider the implications and limitations of applying this normalization to temporally correlated stochastic datasets.

SampEn has been shown to be robust to variations in dataset length [28]. This was observed for the stationary datasets (α < 1; Figure 6A). The precision of SampEn estimates increased for the stationary datasets as *N* increased. By contrast, the distributions of SampEn estimates drifted apart for the nonstationary processes suggesting that greater regularity was detected in processes characterized by the same temporal correlation properties; however, in general, the distribution shapes were similar for increasing dataset lengths.

By contrast, the normalized SampEn estimates produced a distinct pattern of results. The qualitative shape of the SampEn estimates as a function of α were comparable to the unnormalized values. However, normalization produced clear differences in SampEn as a function of dataset length for the stationary datasets, whereas longer datasets were characterized by lower normalized estimates (Figure 6B). This is not surprising, given that longer datasets have larger maximum values; however, it suggests a potential problem because longer datasets with the same temporal correlation properties are characterized by greater regularity. Thus, normalization may not be appropriate for stochastic datasets.

## 4. Discussion

This paper examined considerations for applying entropy methods, specifically SampEn, to temporally correlated stochastic data, which are representative of variables that are often calculated in biomechanical research, such as spatiotemporal gait parameters and center of pressure trajectories. The main problems considered were (1) biases in SampEn in the presence of outliers, (2) the inability of SampEn to distinguish distinct temporal correlation properties, and (3) the limited interoperability of SampEn estimates obtained for datasets of substantially different lengths.

### 4.1. ARFIMA Modeling for Estimation and Classification of Temporally Correlated Stochastic Datasets

Solutions to some of the problems mentioned above relied on the ability to classify stochastic data as stationary or nonstationary. In the fGn/fBm model, stationary and nonstationary processes are related by the Hurst exponent [46]. Stochastic processes can also be quantified on a continuous scale by measuring power law relations of changes in the variance over different time windows (i.e., $SD\left(t\right)\propto t^{\alpha }$) or changes to the spectral power over different frequencies (i.e., $P\left(f\right)\propto f^{-\beta }$). Here, we opted for the scaling exponent, α, which ranges from ]0, 1[ for stationary processes and ]1, 2[ for nonstationary processes. By generating simulated datasets, we were able to provide insights across the range of temporal correlations that are commonly observed in biomechanical variables.

We estimated the temporal correlation properties and classified the simulated datasets using ARFIMA (0, *d*, 0) models. There are several notable benefits to this approach. First, the difference order, *d*, can be directly converted to α, which allows consistent interpretation of the measures obtained from ARFIMA and DFA, which has historically been used in the human movement literature (e.g., [63,64]). Second, ARFIMA models do not require hyperparameter selection—unlike other methods, such as DFA—where multiple decisions must be made to obtain parameter estimates, which reduces the potential for discrepancies across studies. Third, and most importantly, ARFIMA models produce more accurate and precise estimates of temporal correlations compared to more common methods, such as DFA, which contributes to better classification of datasets as stationary or nonstationary.

Accurate and reliable classification of stochastic datasets has been considered previously [57,65]. Here, we demonstrated that data classification is also important for obtaining accurate and reliable SampEn estimates. Data cleaning procedures were more effective, and the interpretability of SampEn estimates were improved when data classification and estimation was conducted using ARFIMA modeling. Moreover, SampEn has trouble distinguishing stationary processes with distinct temporal correlation properties, which suggests the need to leverage multiple analytical techniques to characterize and compare the dynamics of biomechanical datasets, which is discussed in more detail below.

### 4.2. Outlier Removal Reduces Biases in SampEn Estimates

Past work has shown that outliers artificially reduce entropy estimates by inflating the estimated standard deviation relative to the scale of the ‘true’ dynamics [43,44,45], thus artificially inflating the *r* hyperparameter. Here, SampEn estimates were biased by approximately 25% when only 1% of the observations were manipulated. The bias persisted when smaller amplitude outliers were generated in separate analyses not reported here. However, the bias magnitude decreased to approximately 20% and 7% for outliers with 2× and 1× the peak-to-peak amplitude, respectively. Because lower SampEn estimates can result both from increased regularity or increased *r* due to the presence of outliers [45], we recommend that researchers cautiously examine datasets for outliers before applying entropy techniques. Otherwise, the resulting interpretations may be inconsistent and/or incorrect.

We examined several methods for reducing biases, including adjusting the radius of similarity, *r*, and outlier removal that was agnostic of the classification of the dataset as stationary or nonstationary. Both methods were effective at reducing the bias, particularly for stationary datasets. However, positive biases remained for nonstationary datasets with persistent increments. This occurred because outlier detection was less effective for these types of superdiffusive processes.

To address this limitation, we used ARFIMA modeling to classify datasets as nonstationary. Nonstationary datasets were differenced to improve outlier detection. We did not remove the outliers from the differenced datasets and then re-integrate them. Because each difference results from multiple datapoints, removing them can create sizeable discontinuities in the re-integrated dataset, which can also bias SampEn estimates. Instead, outliers were directly removed from the nonstationary datasets. This procedure substantially reduced biases in SampEn across the entire range of the fGn/fBm model (0 < α < 2).

One concern is that outlier removal is itself a methodological procedure that requires parameter tuning and defining what counts as an outlier. We recommend being conservative when data cleaning by restricting removal only to the most extreme observations. If a substantial proportion of observations are flagged as outliers, the detection threshold may be too strict. Unfortunately, we cannot provide a simple recommendation for determining what percentage of the data would represent an unjustifiable number of outliers. Outliers should be rare unless there is suspicion that the data were generated by a distinct mechanism (e.g., the experimenter noticed that a participant was distracted).

Another concern is that outlier removal can create datasets with different numbers of observations. Outlier replacement could reduce data loss and further improve the precision obtained for short datasets. We experimented with several methods, including mean and median replacement, but did not find additional improvements in entropy estimates. More sophisticated procedures have been developed (e.g., [66]) but were not examined. If outlier removal creates substantial variation in dataset length, it may be helpful to crop all datasets to the same length. Fortunately, the accuracy and precision of SampEn have been shown to be relatively robust to variations in sample size [28].

However, we observed that SampEn estimates were robust to differences in dataset length for stationary stochastic processes but not nonstationary stochastic processes (see Table 3 and Figure 6A). We recommend that researchers ensure that their results are not dependent on dataset length, particularly when working with nonstationary datasets. This can be achieved in several ways. First, if sufficient data are collected, then datasets can be cropped. The pattern of results should be consistent across changes in dataset length, even if the estimates begin to depart. Second, additional data can be collected during pilot work and analyses performed to determine the minimum data needed to obtain reliable estimates under specific experimental conditions or from different populations.

Moving forward, the effectiveness of outlier removal procedures needs to be assessed on ‘real-world’ datasets. The simulated datasets examined in this paper are representative of biomechanical datasets obtained from human participants [67,68]. However, in practice, measurements obtained in laboratory or clinical settings will not be as clean and well-behaved, and outliers may not be as easy to detect. While the scaled median absolute deviation threshold was extremely effective for outlier identification and removal, researchers may need to tailor this approach to meet their specific needs.

### 4.3. Interpreting SampEn Estimates from Temporally Correlated Stochastic Data

SampEn is a regularity statistic that quantifies the frequency of repeating patterns in a dataset; but it does not indicate the nature of the observed patterns. This is not a problem when researchers are interested in understanding whether data are more regular under one set of conditions compared to another. However, SampEn may be unable to distinguish between distinct temporal correlation properties when used alone (see Table 3).

Our results highlight a significant limitation of SampEn, namely, the many-to-one correspondence between regularity and temporal correlations for stationary stochastic datasets. Entropy methods, such as ApEn and SampEn, were developed to avoid phase space reconstruction, which requires large amounts of noise-free data to estimate KS entropy [22]. Whereas KS entropy is defined in the limits of *m* →∞, *r* → 0, and *N* →∞, ApEn was developed to conduct a partial characterization using smaller *m* and larger *r* that requires less data and is less sensitive to noise [29]. Revisiting a point made by Pincus [29], failures to distinguish differences in regularity do not indicate that the dynamics on the reconstructed attractor are the same. We reiterate a more general point that similar values of SampEn do not indicate that the dynamics being compared are the same.

We were unable to identify straightforward solutions or transformations to remedy this issue. For example, differencing stationary datasets before submitting them to SampEn produced a monotonic increasing function as α increased from 0.1 to 0.99—see also Figure 2 in [35]. This procedure could be useful if all datasets are stationary, but often experimental studies contain datasets on both sides of the 1/f boundary (e.g., [69]). Indiscriminately applying the differencing procedure would re-introduce the many-to-one mapping problem for nonstationary datasets.

There is the additional problem that the dataset length biases the entropy estimates for nonstationary processes such that longer datasets are characterized by greater regularity despite having the same temporal correlation properties as shorter ones (Figure 6). This occurred even though SampEn does not explicitly depend on *N*. The observation of greater regularity for longer datasets may be attributable to the fact that there are more observations within the radius of similarity, r, such that the conditional probability A/B becomes larger, which would reduce SampEn. The problem is determining which estimates are correct. One solution is to assume that the longer time series produce better estimates. On the other hand, it may be inappropriate to apply SampEn to these types of nonstationary datasets because of the poor precision across estimates. This suggests the need to convert nonstationary processes to stationary processes before applying SampEn. The main limitation of this approach is that it does not solve the many-to-one mapping problem described above. An alternative is to use methods specifically developed to handle nonstationarity, such as Control Entropy (e.g., [31]).

The inability to distinguish stochastic datasets with distinct temporal correlation properties using SampEn creates a problem for researchers and clinicians interested in examining differences in the dynamics of biomechanical variables. There is a greater likelihood of reporting false negatives when experimental conditions or groups are characterized by similar degrees of regularity, even if the underlying dynamics are different. To avoid this, we recommend combining the model-free approach of entropy methods and the model-based approach of ARFIMA modeling to quantify regularity *and* characterize the temporal correlations producing the observed regularities.

### 4.4. Normalization of SampEn Estimates

One goal of this paper was to examine the use of a normalization method to increase the interoperability of SampEn estimates obtained from datasets with different lengths. Experimental designs and conditions vary widely depending on the specific research questions, leading to different quantities of data and hyperparameter selections. This makes it hard to compare entropy estimates because the range of possible values is a function of *m*, τ, and *N* (Equation (3)).

Because SampEn is bounded, the maximum entropy value can be used to normalize values between 0 and 1. This could, in principle, account for differences in dataset length. Other work has found that SampEn is robust across different dataset lengths [28]. Our results confirm this observation but only for the stationary stochastic processes (α < 1). For the nonstationary processes (α > 1), greater regularity was observed for longer datasets. Coupled with the decreased precision observed for the nonstationary datasets, caution should be taken when applying SampEn to nonstationary data, particularly when datasets have different numbers of observations.

One rationale for applying normalization is to increase interoperability. There are multiple ways to increase interoperability, including data transparency, method sensitivity, and parameter normalization. While parameter normalization was not appropriate for stochastic datasets, other opportunities remain. First, studies applying entropy methods should report hyperparameter values *and* how those values were selected. Second, unless there are restrictions from funding sources or other substantial barriers, researchers can make datasets and analysis routines publicly available, which was performed for this paper (see Data Availability Statement). Third, normalization may be plausible when working with experimental datasets characterized by a mixture of deterministic and stochastic components where hyperparameter selection can be guided by techniques adopted from phase space reconstruction, such as average mutual information [61] and false nearest neighbors [70]. This should be explored in future work.

## 5. Conclusions

In summary, this paper has highlighted considerations when applying entropy, specifically SampEn, to stochastic datasets with a range of temporal correlations, which are representative of biomechanical variables. We demonstrated the use of ARFIMA modeling for estimating temporal correlation properties and classifying stochastic datasets as stationary or nonstationary. We then leveraged ARFIMA modeling to improve the effectiveness of data cleaning procedures to mitigate the influence of outliers on SampEn estimates. We also emphasized the limitations of SampEn in distinguishing among stochastic datasets, leading to the suggested use of complementary measures to obtain a more comprehensive understanding of behavioral dynamics. Finally, we demonstrated that the parameter normalization method is not an effective procedure for increasing the interoperability of SampEn estimates for the entirely stochastic datasets examined here.

## Figures and Tables

**Figure 1 entropy-25-00306-f001:**
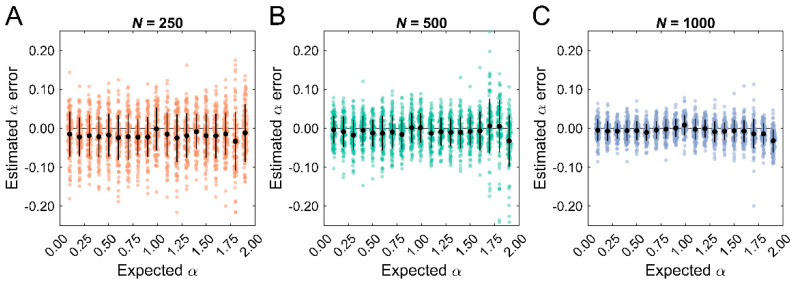
Error in the estimated scaling exponent, α, obtained from the ARFIMA (0, d, 0) model fits. (**A**–**C**). Errors for the short (*N* = 250), medium (*N* = 500), and long (*N* = 1000) datasets, respectively. Each point represents one simulation. The black circles represent the mean error (i.e., bias); error bars represent the standard deviation.

**Figure 2 entropy-25-00306-f002:**
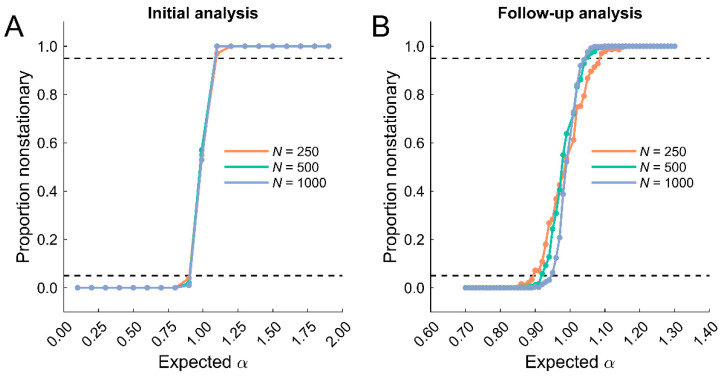
Proportion of simulations classified as nonstationary by the ARFIMA (0, *d*, 0) models. (**A**). The initial analysis with wider gaps between the expected α values indicated that the classification accuracy was close to 50% near the 1/f boundary but less than 5% (dotted black lines) in nearby regions. There were minimal differences across dataset lengths. (**B**). The follow-up analysis with finer gaps revealed that classification accuracy was dependent on dataset length. Short datasets (*N* = 250) were classified least accurately on either side of the 1/f boundary. Medium datasets (*N* = 500) were classified less accurately than long datasets (*N* = 1000) but only below the 1/f boundary.

**Figure 3 entropy-25-00306-f003:**
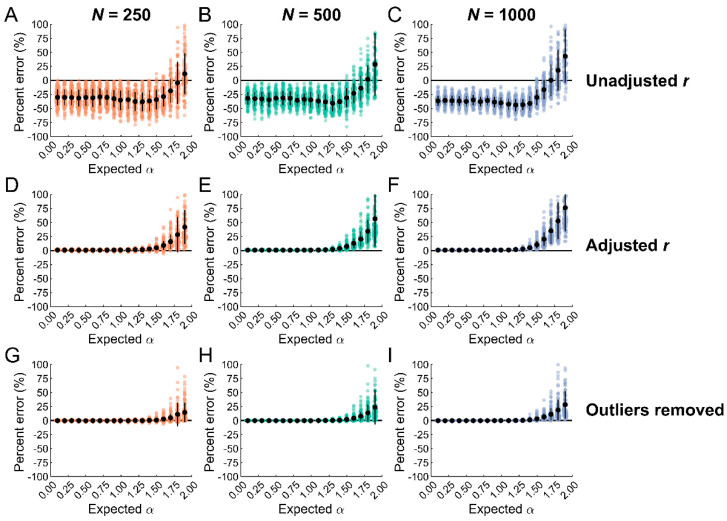
Entropy estimates were biased by small numbers of extreme observations. (**A**–**C**). Biases in entropy estimates, expressed as a percentage, when 1% of the observations were manipulated without adjusting r. Entropy estimates were characterized by negative biases (>25%) except for α > 1.5. (**D**–**F**). Biases in entropy estimates after adjusting r to account for outliers. Near-zero biases were observed for the stationary datasets (α < 1). Biases were less than 5% for nonstationary datasets with anti-persistent increments (1 < α < 1.5). The bias and standard deviation progressively increased for nonstationary datasets with persistent increments (α < 1.5). (**G**–**I**). Biases in entropy estimates after outlier removal with similar results as adjusting r. However, removing outliers further reduced the bias and standard deviation for nonstationary datasets with persistent increments (α < 1.5).

**Figure 4 entropy-25-00306-f004:**
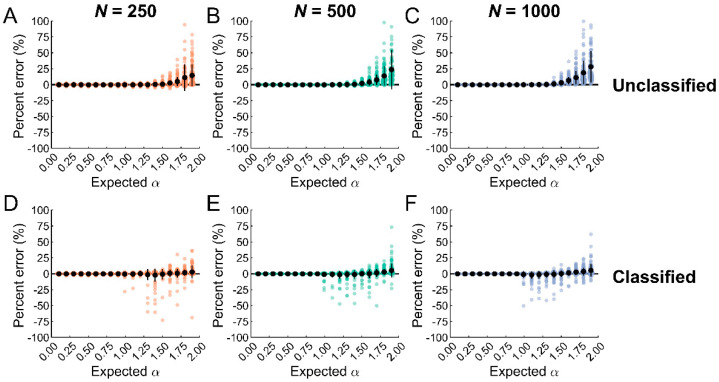
Classification with ARFIMA (0, d, 0) models reduced biases in entropy estimates for nonstationary datasets independent of dataset length. (**A**–**C**). Biases in entropy estimates obtained following outlier removal when classification was not applied. (**D**–**F**). Biases in entropy estimates when outlier removal was preceded by data classification. Data classified as nonstationary were differenced and outliers were identified and removed. This procedure reduced the biases in entropy estimates across the entire α range. However, the standard deviation was inflated for nonstationary datasets with anti-persistent increments (1 < α < 1.5), indicating some tradeoffs with this approach.

**Figure 5 entropy-25-00306-f005:**
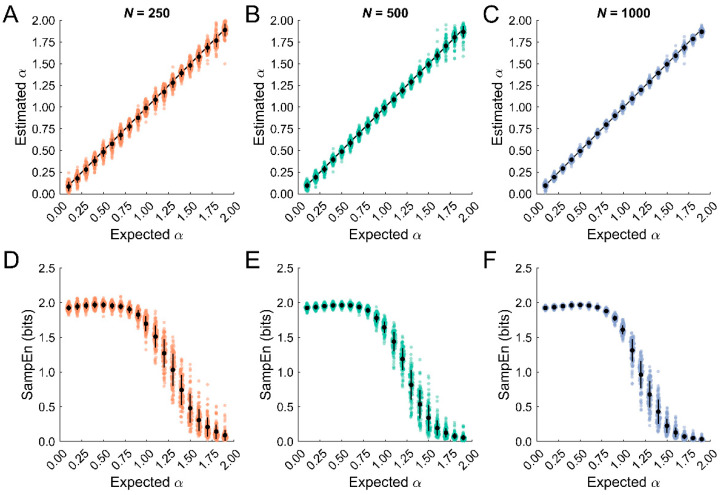
Limitations of SampEn to discriminate temporally correlated stochastic datasets. (**A**–**C**). The estimated and expected temporal correlation properties, measured by the scaling exponent, α. The estimated α values were characterized by a linear relationship with a slope close to 1. Thus, the temporal correlation properties were accurately characterized by ARFIMA modeling independent of dataset length. The black lines represent the expected values. Each data point represents one of 100 simulated datasets. Means are represented by black circles. Error bars are standard deviations. (**D**–**F**). By contrast, SampEn estimates showed a many-to-one correspondence with α, characterized by an inverted parabola that peaked for stationary uncorrelated noise (α = 0.5). Thus, SampEn may not reliably distinguish subtle differences in the dynamics of stochastic data. Each data point represents one of 100 simulated datasets. Means are represented by black circles. Error bars are standard deviations.

**Figure 6 entropy-25-00306-f006:**
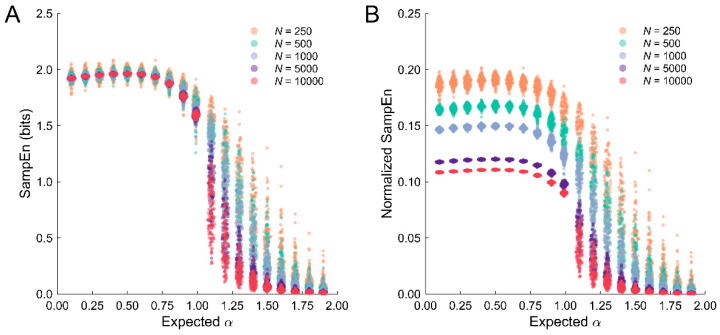
Normalization of SampEn across different dataset lengths. (**A**). Unnormalized SampEn estimates for different values of the scaling exponent, α, and dataset length, N. Each data point represents one simulated dataset. Estimates obtained for the stationary datasets were robust to differences in dataset length, whereas the nonstationary datasets were more sensitive to dataset length. (**B**). Normalized SampEn estimates for the different values of the scaling exponent, α, and dataset length, N. Normalization incorrectly suggests that datasets characterized by the same temporal correlation properties contained different degrees of regularity.

**Table 1 entropy-25-00306-t001:** Parameter values for the simulated data obtained from the ARFIMA (0, *d*, 0) models.

	Parameter	
	Difference Order (*d*)	Scaling Exponent (α)
Stationary (fGn)	−0.4	0.1
	−0.3	0.2
	−0.2	0.3
	−0.1	0.4
	0.0	0.5
	0.1	0.6
	0.2	0.7
	0.3	0.8
	0.4	0.9
	0.49	0.99
Nonstationary (fBm)	0.6	1.1
	0.7	1.2
	0.8	1.3
	0.9	1.4
	1.0	1.5
	1.1	1.6
	1.2	1.7
	1.3	1.8
	1.4	1.9

**Table 2 entropy-25-00306-t002:** Bias and standard deviation (SD) of the estimated scaling exponents, α.

	*N* = 250		*N* = 500		*N* = 1000	
Expected α	Bias	SD	Bias	SD	Bias	SD
0.1	−0.015	0.058	−0.004	0.036	−0.005	0.028
0.2	−0.022	0.050	−0.009	0.039	−0.007	0.025
0.3	−0.019	0.053	−0.017	0.038	−0.007	0.020
0.4	−0.022	0.053	−0.005	0.038	−0.006	0.026
0.5	−0.017	0.055	−0.012	0.034	−0.006	0.024
0.6	−0.024	0.058	−0.013	0.044	−0.010	0.022
0.7	−0.021	0.059	−0.010	0.041	−0.004	0.023
0.8	−0.023	0.047	−0.016	0.037	−0.002	0.025
0.9	−0.022	0.051	0.001	0.038	0.001	0.023
0.99	−0.001	0.055	0.001	0.038	0.008	0.026
1.1	−0.015	0.057	−0.013	0.037	−0.002	0.027
1.2	−0.025	0.061	−0.009	0.036	−0.001	0.024
1.3	−0.019	0.058	−0.010	0.036	−0.009	0.027
1.4	−0.008	0.045	−0.010	0.041	−0.008	0.023
1.5	−0.019	0.056	−0.008	0.037	−0.006	0.029
1.6	−0.019	0.058	−0.006	0.050	−0.007	0.028
1.7	−0.014	0.055	0.006	0.070	−0.014	0.037
1.8	−0.033	0.077	0.005	0.067	−0.014	0.027
1.9	−0.012	0.075	−0.032	0.066	−0.032	0.030

**Table 3 entropy-25-00306-t003:** Mean and standard deviation (SD) of SampEn(*m* = 1, *r* = 0.25, *τ* = 1) for each expected value of the scaling exponent, α, and dataset length, *N*.

	*N* = 250		*N* = 500		*N* = 1000	
Expected α	Mean	SD	Mean	SD	Mean	SD
0.1	1.927	0.043	1.927	0.031	1.926	0.016
0.2	1.947	0.053	1.939	0.030	1.937	0.018
0.3	1.959	0.047	1.953	0.027	1.951	0.017
0.4	1.971	0.050	1.962	0.029	1.964	0.017
0.5	1.969	0.043	1.965	0.024	1.968	0.015
0.6	1.957	0.045	1.962	0.029	1.960	0.018
0.7	1.947	0.046	1.940	0.032	1.934	0.015
0.8	1.906	0.052	1.893	0.039	1.881	0.026
0.9	1.830	0.069	1.781	0.064	1.778	0.040
0.99	1.697	0.109	1.646	0.083	1.613	0.067
1.1	1.512	0.161	1.442	0.141	1.315	0.159
1.2	1.273	0.205	1.189	0.156	0.963	0.196
1.3	1.033	0.234	0.817	0.218	0.680	0.193
1.4	0.744	0.216	0.539	0.204	0.431	0.177
1.5	0.481	0.210	0.342	0.188	0.228	0.100
1.6	0.310	0.159	0.195	0.109	0.131	0.069
1.7	0.211	0.137	0.126	0.068	0.074	0.035
1.8	0.144	0.104	0.079	0.043	0.051	0.026
1.9	0.091	0.080	0.058	0.043	0.035	0.019

## Data Availability

The data, analysis routines, and results presented in this study are openly available at https://doi.org/10.5281/zenodo.7595303.

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
