# Peer review of "Considerations for Applying Entropy Methods to Temporally Correlated Stochastic Datasets"

_entropy, 2023, doi:10.3390/e25020306_

Round 1

Reviewer 1 Report

In the reviewed manuscript, the authors present the results of the research regarding the application of the entropy methods to temporally correlated stochastic datasets. The topic is interesting and actual. However, to my best mind, it is necessary to make some corrections before the manuscript acceptance. Below, i present my remarks. 

1. The Abstract should be extended by adding brief information about the obtained results and an analysis of the obtained results.

2. The paper will look better, if at the end of the Literature review to allocate the unsolved parts of the general problem.

3. Please, at the end of the Introduction section add the main contribution of the authors' research.

4. The theoretical part of the research is absent. Please, add the section Materials and Methods, there present the methodology of the research with the corresponding structural block-chart. Please, in this section, present also the mathematical foundation of the research. In our case, the section Materials should be renamed as Experiment.

5. The Conclusion section also should be extended by adding the more concrete information about obtained results and its analysis. Moreover, add the information concerning the further application of the obtained results.

Reviewer 2 Report

COMMENTS ABOUT

Considerations for applying entropy methods to temporally correlated stochastic datasets

1.      Authors explained that instead of using ApEn they used SaEn, with correct and known arguments, but why not to improve further the already known form of SaEn?

2.      As authors said, ApEn was improved as SaEn, and this one in the form of MSE. But being this true, why not to work with MSE instead of SaEn?

3.      About (2), in lines 108-109, authors strengthen the fact that MSE has been used in references [9-13]. Later, in lines 110-115, SaEn is used in [14-17]. If you are so confident in SaEn instead of MSE, provide more cases (references) of SaEn surpassing MSE in biomedical applications.

4.      In this sense (3), explain why SaEn is still better than MSE up to this point (at least) in references [9-17].

5.      Although in lines 117-119, authors say they focus on temporal correlated stochastic data, they use spatiotemporal data. How to “retrieve” spatial information? At least, as you describe this, space information is lost.

6.      As authors claim, recommendations for handling outliers when working with stochastic datasets are lacking. Authors pose the question: Should r be adjusted to reduce the bias? I ask authors: should you adjust “r” and “m” at the same time?

7.      Could (6) help to strike a balance between minimizing bias and preserving the underlying dynamics?

8.      Lines 157-160. I’m not completely agree with this argument. Remember that there do exist constraints in the length of the time series. There are versions of entropy for short and long data. Defining long and short is another issue. How to encompass all of this in order provide a clear and true argument here?

9.      Lines 168-176. Authors say that: “Specifically, we examine the use of autoregressive fractionally integrated moving average (ARFIMA) modeling to estimate the temporal correlation properties of stochastic datasets …”. There are other ways to do this, for instance, Permutation Entropy, Ordinal Probabilities and Machine Learning. Please add a section where you explain the advantages (if any) of using ARFIMA versus (AT LEAST) those ones. From the context,  it is obscure the way ARFIMA suddenly shows up.  

10.   The same for classifying datasets as stationary or nonstationary.

11.  Remarks (8) and (9) just partly done in lines 179-185.

12.  Lines 177-228. Agree.

13.  From lines 250-269, remarks done in (1)-(4) are a bit justified.

14.  Lines 277-293. Recall remark (8). At least in the way it is written, you contradict that SaEn does not depend of N (I’m still not agree with this because as you know, SaEn=SaEn(m,r,N)).

15.  Line 291. According to general literature, all your N are classified as short data.

16.  Section 3. Considering all (1-15) above, I’m basically agree here. However, there are alternatives to the way authors present their study. For instance, about entropy in stationary and non-stationary datasets, there do exist entropies for non-stationary datasets. Explain differences.

17.  As far as I could see, authors claim that their method is applicable to biomedical time series, and then authors give examples of them but authors never use any biomedical datasets to test their method. Not even one.

18.  About (17), authors provide a lot of biomedical applications references, but as mentioned, no biomed application is developed.

19.  As a consequence of (17), there are no biomedical interpretation of the method.

20.  Authors use Hurst exponent. It is known that it can also be used in long-range dependence. Could your results deal with this?

21. Authors frequently mention that thier method is applicable to biomedical datasets without developing  anyone. I recommend to include at least one case of this. 

22. It seems that authors were forced to use SaEn because they use SaEn=SaEn(m,r,N,tau) in spite of knowing that MSE (for instance) has proved to be better tan SaEn. Explain from the beginning also.

Round 2

Reviewer 1 Report

Firstly, I would like to remark on the incorrectness of the author's responses to the reviewer's remarks. The reviewers' remarks focused on improving the paper's structure and content. And the remark that the reviewers do not read the manuscript is not correct. 

2-3 answers. I have read the Problem statement. I want to note that unsolved parts of the general problem and Problem Statement differ from each other. Moreover, the Introduction section should contain the actuality of the problem, its importance, existing ways of problem-solving and the authors' proposition in this subject area. At the end of the Introduction, it will be better to add the main contribution of the authors' research.  The next section, a Literature review, should contain a critical analysis of the current research in this subject area with an allocation of advantages and disadvantages of the appropriate solution. At the end of this section, it's necessary to allocate the unsolved part of the general problem with the following Problem Statement.    

Regarding the fourth answer. The research article should be structured correctly. The readers should see the theoretical foundations of the research, the experimental part, the results and the discussion of the obtained results. In your case, you have the computational simulation. This is a type of experiment.  I saw that you had applied the ARFIMA models. However, any simulation contains its methodology and step-by-step procedure. In your case, I see a mix of theoretical and practical parts. Moreover, I did not ask before about the method of Shannon entropy calculation. There are various methods—for example, maximal likelihood, James and Stein, Bajes, etc. Please, explain your choice. Did you compare the various methods of Shannon entropy calculation?   Thus, the manuscript should be reformatted for better understanding. 

The remark regarding the Conclusion is the same.